# SIR-SI model with a Gaussian transmission rate: Understanding the dynamics of dengue outbreaks in Lima, Peru

**Max Carlos Ramírez-Soto**[1]*, **Juan Vicente Bogado Machuca**[2]°, **Diego H. Stalder**[3]°, **Denisse Champin**[1], **Maria G. Mártinez-Fernández**[3], **Christian E. Schaerer**[2]

**1** Facultad de Ciencias de la Salud, Universidad Tecnologica del Peru, Lima, Peru, **2** Polytechnic School, National University of Asuncion, San Lorenzo, Paraguay, **3** Faculty of Engineering, National University of Asuncion, San Lorenzo, Paraguay

° These authors contributed equally to this work.
* maxcrs22@gmail.com, c20330@utp.edu.pe

## Abstract

### Introduction

Dengue is transmitted by the *Aedes aegypti* mosquito as a vector, and a recent outbreak was reported in several districts of Lima, Peru. We conducted a modeling study to explain the transmission dynamics of dengue in three of these districts according to the demographics and climatology.

### Methodology

We used the weekly distribution of dengue cases in the Comas, Lurigancho, and Puente Piedra districts, as well as the temperature data to investigate the transmission dynamics. We used maximum likelihood minimization and the human susceptible-infected-recovered and vector susceptible-infected (SIR-SI) model with a Gaussian function for the infectious rate to consider external non-modeled variables.

### Results/principal findings

We found that the adjusted SIR-SI model with the Gaussian transmission rate (for modelling the exogenous variables) captured the behavior of the dengue outbreak in the selected districts. The model explained that the transmission behavior had a strong dependence on the weather, cultural, and demographic variables while other variables determined the start of the outbreak.

### Conclusion/significance

The experimental results showed good agreement with the data and model results when a Bayesian-Gaussian transmission rate was employed. The effect of weather was also observed, and a strong qualitative relationship was obtained between the transmission rate and computed effective reproduction number $R_t$.

**Data Availability Statement:** Data are freely available and can be accessed from CDC Peru

(https://www.dge.gob.pe/salasituacional/sala/index/salasit_dash/143).

**Funding:** This study was supported by the Universidad Tecnologica del Peru (Grant reference No. P-2020-LIM-01; grant recipient: Max Carlos Ramírez-Soto) and the PRONII - PROCIENCIA - CONACYT – FEEI, Paraguay (Grant recipient: Dr. Christian E. Schaerer and Dr. Diego H. Stalder). The funders had no role in the study design, data collection and analysis, decision to publish or the preparation of the manuscript.

**Competing interests:** The authors have declared that no competing interests exist.

# 1 Introduction

Dengue is a viral disease that can result in hospitalization and even death, and its main transmission vector is the *Aedes aegypti* mosquito [1]. Its endemic characteristics make it a public health problem. Dengue is now endemic in Africa, America, Asia, and the Western Pacific [2, 3], and South America has seen a dramatic increase in cases in countries such as Colombia, Ecuador, Paraguay, Peru, Venezuela, and Brazil [4]. In Peru, dengue cases are widely distributed in different geographical regions such as the coast, mountains, and jungle.

Between 2004 and 2017, there was an increase in dengue cases in Lima. The districts with the highest incidence rates were Comas (+30 cases/100,000 inhabitants), Lurigancho-Chosica, and Puente Piedra (10–29 cases/100,000 inhabitants) [2, 5]. Despite the high incidence of dengue and the spread of *Ae. aegypti* in more than 10 districts of Lima [6–9], dengue surveillance and prevention strategies have been limited to searching for febrile patients, environmental and hygiene education, and surveillance of entomological indicators (e.g., larva, pupa, container, and aedic indices) [10]. Recent studies in Peru have shown that the entomological indicators of *Ae. aegypti* calculated from epidemiological surveillance have limited utility in detecting high-risk areas or populations for dengue infection [5, 8, 10]. The presence of dengue in Lima is related to the growth of the city, the inadequate provision of drinking water services, intra-domicile and community water storage, inadequate waste disposal, the presence of the transmission vector, and import of cases from other regions. Thus, the entomological surveillance of *Ae. aegypti* is being strengthened to control the transmission of dengue in Lima. Other studies have suggested that the persistence of dengue in Lima can be attributed to insufficient access to essential sanitation services, discontinuous preventive activities, scarce health personnel, and poor community participation in dengue prevention [6, 9, 11, 12].

Several compartmental models have been developed to explain the spread of dengue, from simple models such as susceptible-infected-recovered (SIR) to more complex models such as susceptible-exposed-infected-recovered (SEIR) and human susceptible-infected-recovered and vector susceptible-infected (SIR-SI) [11, 13–15]. These epidemiological models are important for explaining the transmission dynamics of dengue and for developing control measures [16, 17]. However, few studies have modeled dengue transmission dynamics in the context of Peru. Chowell et al. [6] estimated the transmissibility of dengue outbreaks by using the local reproduction numbers and assessed the outbreak dependence on community size as a function of the geographic region. Their findings suggested a hierarchy of transmission events during the significant 2000–2001 epidemic from large to small population areas when the serotypes DEN-3 and DEN-4 were first identified (Spearman $\rho = 0.43$, $P = 0.03$). In another study, Chowell et al. [7] investigated the association between dengue incidence in 1994–2008 and the demographic and climatic factors across geographic regions in Peru. They found that dengue is persistent in jungle areas, where epidemics peak most frequently around March when rainfall is abundant. Differences in the timing of dengue epidemics in the jungle and coastal regions showed significant correlations with the seasonal temperature cycle. Despite these findings, more studies are needed to explain the dengue transmission dynamics in low-transmission areas such as Lima, Peru.

The objective of this study was to describe the transmission dynamics of dengue in three districts of Lima where *Ae. aegypti* is circulating by using the SIR-SI model for the period of 2016–2020. We performed a correlation analysis between dengue cases and climatological variables. In this paper, we present the SIR-SI model used to determine the transmission rate depending on temperature and a Gaussian function for non-modeled variables. We also discuss the parameter estimation method and the model selection criteria.

## 2 Materials and methods

In this study, we used the SIR-SI model incorporating climatic variables as proposed by Lee et al. [18] to fit the epidemiological curve to data on recurrent outbreaks of dengue in Lima. Then, we used the differential evolution algorithm [19] to fit temperature-based variables inside the SIR-SI model as a Gaussian function. We then selected the most appropriate model based on several metrics. Experiments were performed to evaluate different methods and criteria.

### 2.1 Study-area dataset

Weekly cases of dengue, which were organized for 43 districts of Lima Province, were obtained from the National Center for Epidemiology, Disease Prevention and Control (CDC Peru) [20]. We added all weekly cases from all districts to obtain the number of outbreaks in Lima for 2017, 2019, and 2020. We focused on the period between January 1, 2017, and December 31, 2020. This study received approval from the Ethics Committee at Universidad Tecnológica del Peru. Because the number of cases is low and the population size of the district is limited, making a daily distribution of the time series would be very noisy, because is possible to have unreported cases. In addition, the time period between the occurrence of cases and registration in the notification system may have delays. So, we restrict our study to a weekly scale.

As shown in Fig 1, the study considered three districts of Lima: Comas, Puente Piedra, and Lurigancho. Comas is situated in the north of Lima and is bound by San Juan de Lurigancho to the east and Puente Piedra to the west. The altitude varies between 150 and 811 m. Comas has an area of 48.75 km$^2$ and a population density of 10,813.6 inhabitants/km$^2$. The population

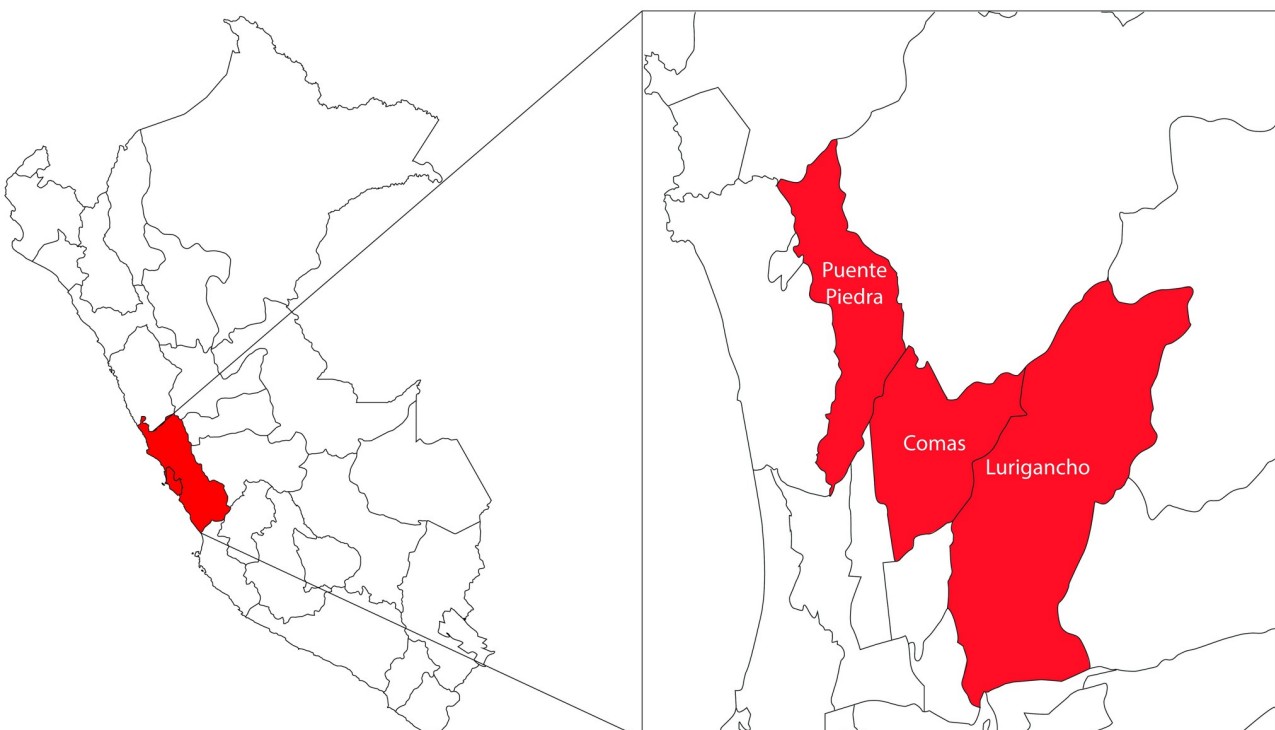

**Fig 1. Geographic location of the selected districts in Lima, Peru.**

was 524,894 inhabitants in 2017 according to the National Census of the National Institute of Statistics and Informatics (INEI). Comas has an arid subtropical climate, and it is hot in summer and mild in winter. The minimum average temperature is 14.2˚C, and the average maximum temperature is 24.5˚C (average 22.1˚C). Lurigancho has an area of 236.47 km$^2$ and an altitude of 850 m. The estimated population is 358,754 inhabitants. The weather is sunny almost all year round, but sporadic rainfall occurs between December and March because of its proximity to the mountains [21].

## 2.2 Climatic data

Temperature data for the three districts were obtained from the Global Data Assimilation System (GDAS). This system integrates observations from weather stations to make forecasts from the global models of the National Oceanic and Atmospheric Administration (NOAA, https://www.noaa.gov/climate). GDAS combines the observations into a three-dimensional model space that includes surface observations, balloon data, wind profile data, aircraft reports, buoy observations, radar observations, and satellite observations. Weekly data were obtained for each year. For each week, the maximum, minimum, and average temperatures were obtained.

Fig 2(A) shows the minimum, maximum, and average weekly temperature distributions at the geographic center of Lima between 2015 and 2020. The average temperature oscillates between 16˚C and 27˚C. The maximum temperature of 31.30˚C occurs in February.

Fig 2(B) superposes the temperature distribution of each epidemiological week for all considered years and districts. The temperature in Lima is slightly higher (by 2˚C) than that of the analyzed districts. The maximum temperature was observed between epidemiological weeks 7 and 11. The minimum temperature was observed between epidemiological weeks 29 and 35.

There is significant literature about the influence of climatic factors such as temperature and precipitation on the life cycle of mosquitoes. Several studies in different countries have stated that a lagged temperature effect may explain dengue variability [22].

We performed a cross-correlation analysis on the temperature and dengue to obtain the lag lead between the two time series and determine the overall correlation between the dengue incidence rate and mean temperature during the study period. We used the Pearson cross-correlation method [23]:

$$\rho_{xy}(\tau) = \frac{C_{xy}(\tau)}{\sqrt{C_{xx}(0)C_{yy}(0)}}, \qquad (1)$$

where $C_{xy}(\tau) = E[x(t) - u_x][y(t + \tau) - u_y]$ and $x(t)$ and $y(t + \tau)$ are time series of dengue cases and the mean temperature (lagged by $\tau$ time steps), respectively. $u_x$ and $u_y$ are the mean values of $x(t)$ and $y(t)$, respectively.

## 2.3 Temperature-dependent SIR-SI model with exogenous variables

The model was built under the following assumptions.

**Assumption 1**. *The human and mosquito populations are mixed homogeneously. Each mosquito has an equal probability of biting a given human.*

**Assumption 2**. *In an outbreak, cases are a small fraction of the total population. Hence, only one strain serotype was considered for all outbreaks.*

**Assumption 3**. *The period of an outbreak is relatively short. Hence, we did not consider the birth and death of humans due to natural causes and other diseases.*

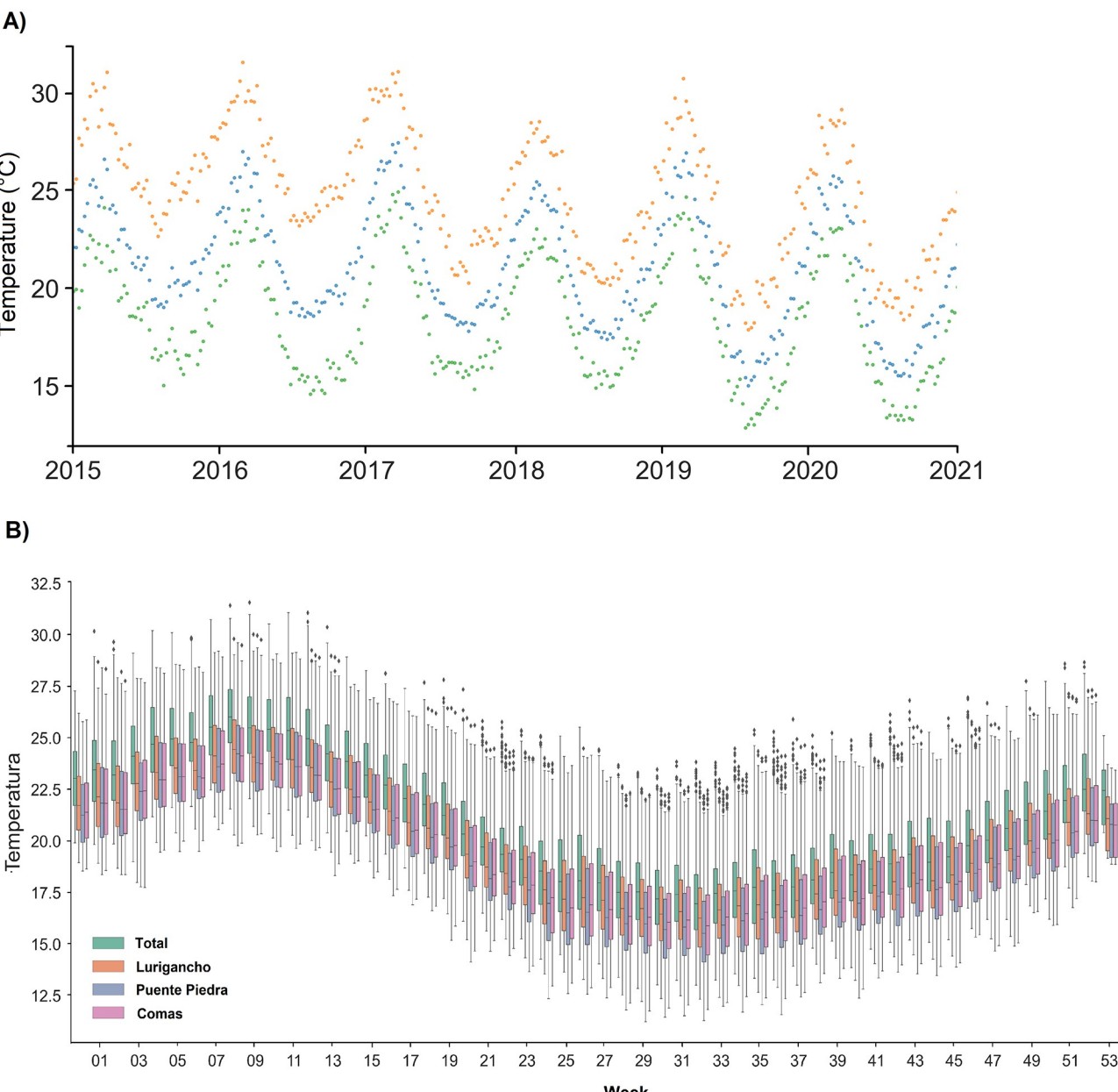

**Fig 2.** Temperature distribution between 2015 and 2020 in Lima, Peru: (A) Weekly minimum (green), maximum (orange), and average (blue) temperatures between 2015 and 2020. (B) Box plots showing the temperature distribution for each epidemiological week and for each district.

**Assumption 4**. *There are no infections of travelers. We considered the mortality of susceptible and infected mosquitoes in both the susceptible and infected compartments, and dependent on the temperature.*

**Assumption 5**. *The population of susceptible humans is limited by the radius of action of the mosquito, so we leave it as a parameter to be estimated.*

**Assumption 6**. *Since we don't have records of the entomological surveillance data, we assume that the mosquito initial population is always double the initial susceptible population of humans, as our baseline work* [18]

Assumption 2 is realistic because relatively few people were infected compared to the entire population of the districts considered in this study. In addition, because only the number of cases was counted, they were considered to be from a single circulating serotype. Assumption 3 establishes that the dynamics of dengue transmission is faster than the dynamics of the births and deaths of the population. The deaths suffered by the mosquito population during an outbreak were attributed to the temperature variations (Assumption 4)

We applied a compartmental model where the host and vector populations are divided into classes. One individual of each population passes from one class to another at a suitable rate set by the model. The SIR-SI model is given by the following equations [24–26]:

$$\frac{dS_h}{dt} = -\beta_{vh}\frac{S_h}{N_h}I_v \tag{2}$$

$$\frac{dI_h}{dt} = \beta_{vh}\frac{S_h}{N_h}I_v - \gamma I_h \tag{3}$$

$$\frac{dR_h}{dt} = \gamma I_h \tag{4}$$

$$\frac{dS_v}{dt} = -\beta_{hv}S_v\frac{I_h}{N_h} - \mu S_v \tag{5}$$

$$\frac{dI_v}{dt} = \beta_{hv}S_v\frac{I_h}{N_h} - \mu I_v, \tag{6}$$

where the sub-indices $h$ and $v$ denote the host and vector, respectively. The parameter $\beta \in \mathbb{R}_+$ is the transmission rate (host-to-vector: $\beta_{hv} \in \mathbb{R}_+$, vector-to-host: $\beta_{vh} \in \mathbb{R}_+$). $\gamma \in \mathbb{R}_+$ is the recovery rate for hosts, $\mu \in \mathbb{R}_+$ is the mortality rate of adults in the vector population, and $S \in \mathbb{N}$, $I \in \mathbb{N}$, and $R \in \mathbb{N}$ represent the susceptible, infected, and recovered fractions, respectively, of a population.

The initial values for the host population are

$$S_{h_0} = 1 - I_{h_0}, \ I_{h_0} = 1/N_h, \text{and } R_{h_0} = 0 \tag{7}$$

The initial values for the vector population are

$$S_{v_0} = 1 - I_{v_0} \text{ and } I_{v_0} = 1/N_v, \tag{8}$$

where $N_h$ and $N_v$ are the host and vector populations, respectively. According to the reference model, $N_v = 2N_h$, and $N_h$ is an estimated parameter. Fig 3 shows a conceptual diagram of the SIR-SI model.

According to Lee et al. [18], the temperature can be incorporated into calculating the transmission rates $\beta_{vh}, \beta_{hv} \in \mathbb{R}$ as follows:

$$\beta_{vh} = x_1 b b_h, \tag{9}$$

$$\beta_{hv} = x_2 b b_v, \tag{10}$$

where $b, b_h, b_v \in \mathbb{R}_+$ are the daily biting rate of a mosquito, the probability of infection (human to mosquito) per bite, and the probability of infection (mosquito to human) per bite,

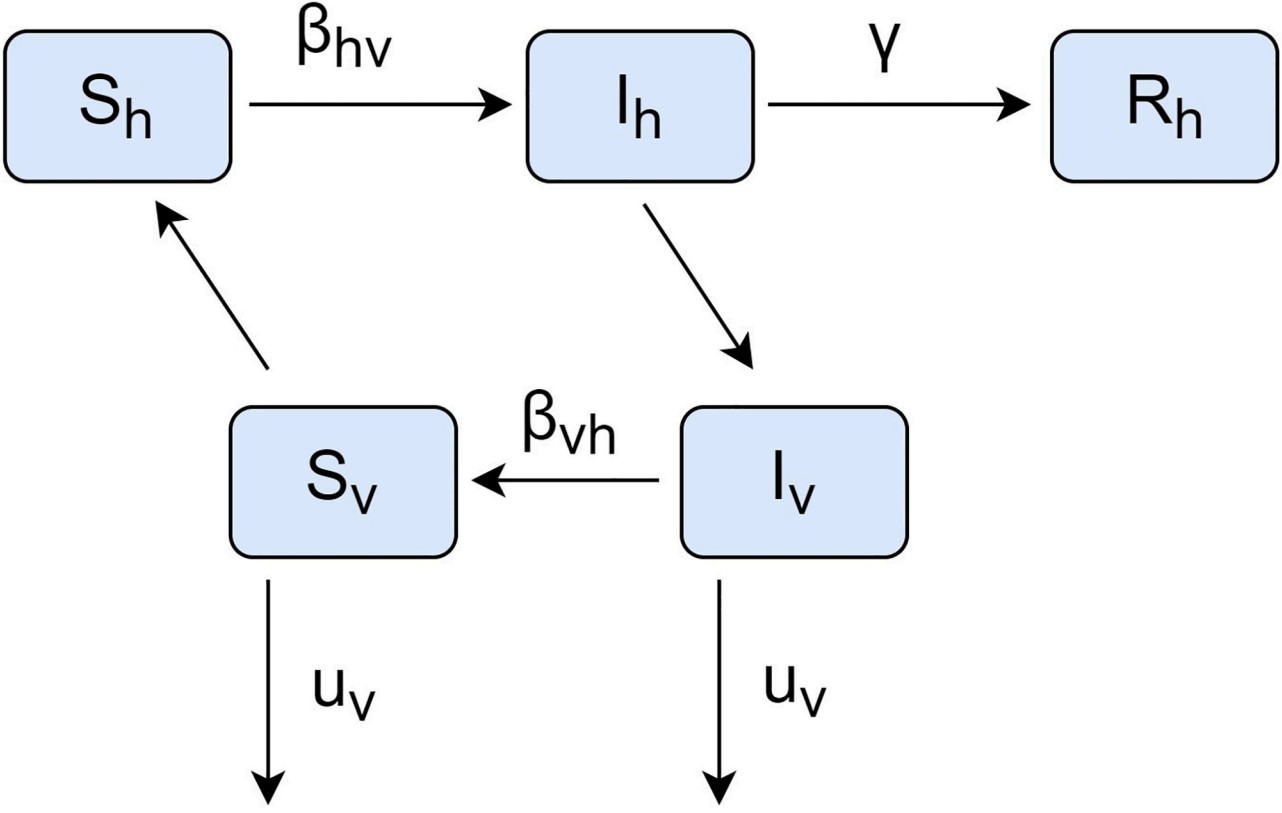

**Fig 3. Conceptual diagram of the SIR-SI model.** *S*: susceptible; *I*: infected/infectious; *R*: recovered; *v*: vector; $\beta_{vh}$: vector-to-host transmission rate; $\beta_{hv}$: host-to-vector transmission rate; $\gamma$: recovery rate; $\mu_v$: vector mortality rate.

respectively. The transmission probabilities $0 \leq x_1 \leq 1$ and $0 \leq x_2 \leq 1$ are constants that can be obtained by data fitting.

The functions $b$, $b_h$, and $b_v$ are temperature-dependent variables whose functions are given in Fig 4 (S1 File). They are defined as follows [27–29] (S1 File).

A lagged cross-correlation can be found between dengue cases and the season. To capture this phenomenon, we considered the effects of weather on the transmission rate by replacing the constants $x_1$ and $x_2$ with a time-dependent variable $\beta_{ex} \in \mathbb{R}$ resulting from a Gaussian function. The Gaussian function is defined as [14]

$$\beta_{ex} := k\,e^{-\frac{(x-u)^2}{2\sigma^2}}, \tag{11}$$

where $k \in \mathbb{R}$ is a constant, $u \in \mathbb{R}$ is the mean, and $\sigma^2 \in \mathbb{R}$ the variance.

Then, $\beta_{hv}$ and $\beta_{vh}$ can be computed as

$$\beta_{vh} = \beta_{ex}\,b\,b_h, \tag{12}$$

$$\beta_{hv} = \beta_{ex}\,b\,b_v, \tag{13}$$

where $\beta_{ex}$ is the value corresponding to the fitted Gaussian function defined in (11). The parameter $\beta_{ex}$ allows us to consider a possible dispersion of cases due to several exogenous factors, including mosquito diapause.

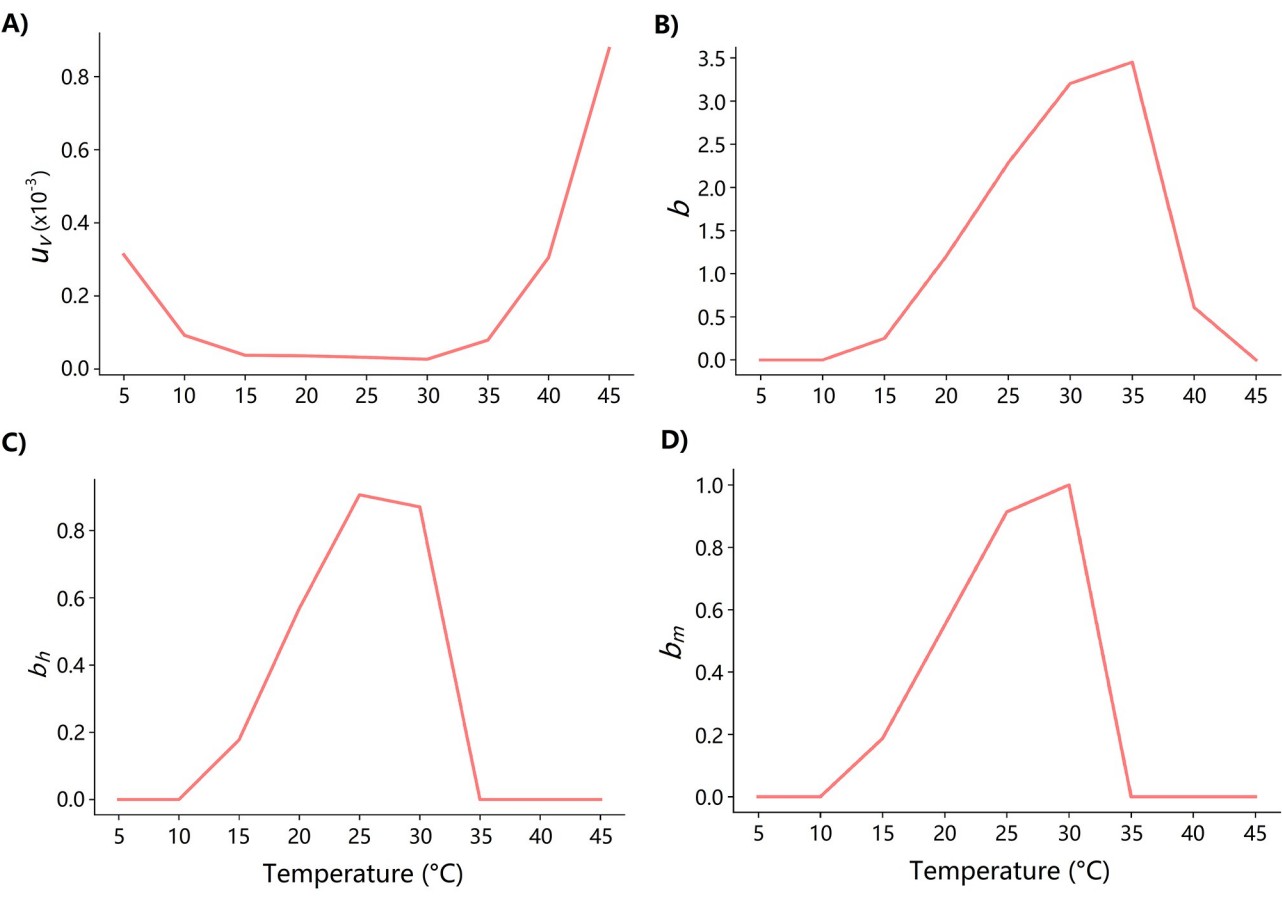

**Fig 4. Temperature-based functions for (a) $\mu$, (b) $b$, (c) $b_h$, and (d) $b_m$.**

## 2.4 Parameter estimation

Parameter estimation can be formulated as an optimization problem, where the best model parameters within the permissible range are found by maximizing a likelihood function [27]. Let $Y$ be a set of weekly reported cases $Y = [y_1, y_2, \cdots, y_n]^T$ during an outbreak containing $n$ consecutive observations. One common assumption to fit the model to given data is that the observational errors follow a normal distribution or likelihood function, such as the least-squares error. If the distribution of each model parameter can be organized in a vector $\theta = [N_h, k, u, \sigma]^T$ so that $\theta \in \Theta$, where $\Theta$ is the parameter space, then let $m_i$ be the prediction of the model for each observation (i.e., a function of $\theta$). Then the likelihood function takes the following form:

$$\mathcal{L}_i(y_i, \theta) := \mathcal{N}(y_i - m_i(\theta), \sigma_i^2), \tag{14}$$

where $\sigma_i(t)^2$ is the assumed variance of the model error (i.e., $1/\sqrt{y_k}$) and $y_k$ is the mean number of cases reported that month. Another assumption is that each observation is statistically independent, so

$$\mathcal{L}(Y, \theta) := \prod_{i=1}^{n} \mathcal{N}(y_i - m_i(\theta), \sigma_i^2). \tag{15}$$

Thus, the maximum likelihood estimation (MLE) takes the following form:

$$MLE := \arg\max_\theta \mathcal{L}(Y, \theta) = \arg\max_\theta \prod_{i=1}^{n} \mathcal{N}(y_i - m_i(\theta), \sigma_i^2)). \tag{16}$$

It is often convenient to work with the natural logarithm of the likelihood so that we can use minimization algorithms. Thus, the problem is equivalent to minimizing the sum of the negative log-likelihood (SNLL):

$$MLE = \arg\min_\theta SNLL := \arg\min_\theta -\sum_{i=1}^{n} \log\left( \frac{1}{\sqrt{2\pi}} e^{-\frac{(y_i - m_i(\theta))^2}{2\sigma_i^2}} \right). \tag{17}$$

Note that, to obtain the model predictions $m_i(\theta)$, the ordinary differential equation should be approximated numerically for each step of the optimization algorithm. Is also important to remark that we fit the data only when the mean reported cases of the month is larger than two, this is to avoid the weeks the reported cases are too low. In addition, note that the free parameters inside $\theta$ define the population size $N_h$ and the Gaussian parameters of the exogenous parameter $\beta_{ex}$ (i.e. $u$, $\sigma$, and $k$).

To solve the optimization problem, we have to use the differential evolution algorithm [30], which can search large areas of parameter space but often requires more function evaluations than conventional gradient-based techniques.

We performed experiments to select the best model and to determine which parameters can be left free to be estimated. Four experiments were set up:

1. Model 1: $u$ estimated, $\sigma = 1$, $k = 1$;

2. Model 2: $u$ estimated, $\sigma$ estimated, $k = 1$;

3. Model 3: $u$ estimated, $\sigma = 1$, $k$ estimated;

4. Model 4: $u$ estimated, $\sigma$ estimated, and $k$ estimated;

The model performances were evaluated for fit to the observed data according to the following criteria.

1. Maximum Likelihood Estimation (*MLE*): The observations were assumed to have a normal error, as described in (16).

2. Akaike Information Criteria (*AIC*) [31, 32]: This measures the relative quality of a statistical model for a given dataset. *AIC* is defined as

$$AIC := 2q - 2\ln(MLE) \tag{18}$$

where $q$ is the number of model parameters and *MLE* is the previously defined MLE.

3. Bayesian Information Criteria (*BIC*) [23]: This is based on the probability function and is closely related to *AIC*. It is defined as

$$BIC := q\ln(n) - 2\ln(MLE) \tag{19}$$

where $q$ is the number of model parameters, $n$ is the number of points evaluated and *MLE* is the previously defined MLE.

Each model was evaluated according to *AIC*, *BIC*, and *MLE*. These metrics were used to assess the quality of the fit [33]. Lower values indicated a better fit.

**Table 1. Model parameters.**

| Parameter | Symbol | Value | Range | Source |
|---|---|---|---|---|
| Infectious period for humans | $\gamma$ | 1 | Fixed | [34] |
| Initial number of humans | $N_{h0}$ | Variable according to each district | | Estimated |
| Initial number of mosquitoes | $N_{v0}$ | $2N_{h0}$ | | [34] |
| Biting rate | $b$ | Temperature-dependent | See Fig 4 | [18] |
| Transmission probability per bite (vector-to-host) | $b_h$ | Temperature-dependent | See Fig 4 | [18] |
| Transmission probability per bite (host-to-vector) | $b_v$ | Temperature-dependent | See Fig 4 | [18] |
| Mortality rate of mosquitoes | $\mu$ | Temperature-dependent | See Fig 4 | [18] |
| Exogenous factors for outbreak | $\beta_{ex}$ | Estimated | See Fig 7 | Defined in (11) |
| Transmissible rate (vector-to-host) | $\beta_{vh}$ | $\beta_{ex}bb_h$ | See Fig 7 | Defined in (12) |
| Transmissible rate (host-to-vector) | $\beta_{hv}$ | $\beta_{ex}bb_v$ | See Fig 7 | Defined in (13) |

The model parameters used in the experiments are listed in Table 1.

During the analyzed period (2016–2020), 650 dengue cases were reported in the three districts. Most of the cases were reported in Comas (76.5%), followed by Lurigancho (13.4%) and Puente Piedra (10.2%). The distributions of cases by week and year are presented in Table 2.

## 3 Results

### 3.1 Cross-correlation analysis

As shown in Fig 5, we found a lagged cross-correlation for the reported dengue cases in each district and each year under study. The maximum values in Fig 5A, 5C and 5E correspond to a 15-week delay between the peaks of the dengue cases and summer temperature while Fig 5B and 5F indicate a slightly shorter delay. Puente de Piedra in 2020 showed a much larger lag of 24 weeks, as shown in Fig 5D. The different lag values indicate that other exogenous factors related to the human population or environment may have had an effect. This was why we needed to include a new parameter to quantify this delay in the model.

### 3.2 Model selection

Table 3 presents the experimental results for the model selection. Model 4 performed better than models 1–4 according to all of the evaluation criteria: *AIC*, *BIC*, and *MLE*. In some cases, it performed up to twice as well as the other models despite being more complex in terms of the number of parameters.

Models 2 and 3 each used $k = 1$ and $\sigma = 1$ while Model 1 only adjusted $u$. These three models performed similarly, which tells us that using fixed parameters degraded the model fit.

Table 4 presents the values of the adjusted parameters $k$, $u$, and $\sigma$ for each model. We can interpret these values in the context of an epidemiological outbreak because $\beta_{vh,hv}$ represents the transmission rate and the parameters $u$, $\sigma$, and $k$ of $\beta_{ex}$ affect it directly.

**Table 2. Reported cases of dengue from 2016 to 2021 (unit: Cases per year).**

| District | 2015 | 2016 | 2017 | 2018 | 2019 | 2020 |
|---|---|---|---|---|---|---|
| Comas | 0 | 48 | 220 | 7 | 2 | 220 |
| Puente Piedra | 8 | 0 | 48 | 1 | 0 | 9 |
| Lurigancho | 0 | 0 | 43 | 0 | 44 | 0 |
| Total | 8 | 48 | 311 | 8 | 46 | 229 |

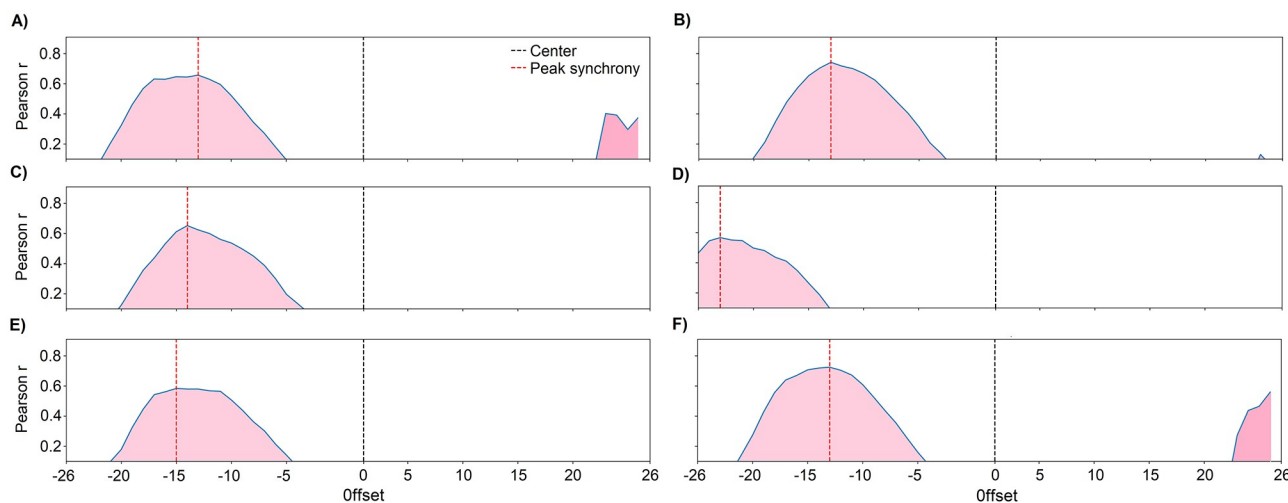

**Fig 5.** Cross-correlation between temperature and cases: Comas in (A) 2017 and (B) 2020, Lurigancho in (C) 2017 and (D) 2019, Puente de Piedra in (E) 2020, and (F) the total. The vertical red line indicates the point of maximum correlation.

The parameters $u$, $\sigma$, and $k$ can be used to characterize outbreaks in addition to the information already obtained with the SIR-SI model and the information from climate-dependent variables.

Fig 6 shows the infectious curves of the model where $x_1$ and $x_2$ are adjusted according to (9) and (9) and the curve obtained by using $\beta_{ex}$ and fitting according to model 4. $\beta_{ex}$ helped model the infectious curve to fit the data more effectively. In contrast, modeling $x_1$ and $x_2$ as constants prevented the effects of non-modeled dynamics such as the weather and diapause to be considered.

Fig 7 shows the values of $\beta_{hv}$ of the benchmark model, $\beta_{hv}$ with $\beta_{ex}$ (proposed in this study), and $bb_v$ (for reference). Observe the action of the $\beta_{ex}$ consists in scaling and introduce a lag in the values of $bb_v$. In essence, model 4 adjusts the three parameters of the Gaussian function

**Table 3. Evaluation of the model fits using *AIC*, *BIC*, and *MLE*.**

| Model | Metric | 2017 | 2019 | 2020 |
|---|---|---|---|---|
| Model 1 | AIC | 1463.0973 | 2690.7484 | 1438.6672 |
|  | BIC | 1463.3999 | 2691.0511 | 1438.9698 |
|  | MLE | 730.5486 | 1344.3742 | 718.3336 |
| Model 2 | AIC | 1438.3085 | 2248.1844 | 1439.1038 |
|  | BIC | 1438.9136 | 2248.7896 | 1439.7089 |
|  | MLE | 717.1542 | 1122.0922 | 717.5519 |
| Model 3 | AIC | 1466.7228 | 1952.3625 | 1424.4774 |
|  | BIC | 1467.3279 | 1952.9676 | 1425.0826 |
|  | MLE | 731.3614 | 974.1812 | 710.2387 |
| Model 4 | AIC | 286.9677* | 1201.5911* | 263.0954* |
|  | BIC | 287.8754* | 1202.4988* | 264.0032* |
|  | MLE | 140.4838* | 597.7955* | 128.5477* |

Abbreviations: AIC, Akaike Information Criterion; BIC, Bayes Information Criterion; MLE, Maximum Likelihood Estimation.

Superscript "*" denotes the best model.

**Table 4. Values of the adjusted parameters for each model in the experiments.**

| Model | Parameter | 2017 | 2019 | 2020 |
|---|---|---|---|---|
| Model 1 | $u$ | 5.4881 | 6.9874 | 5.2181 |
| Model 2 | $u$ | 6.1528 | 7.3931 | 5.1911 |
| | $\sigma$ | 1.0719 | 0.7461 | 0.9843 |
| Model 3 | $u$ | 5.4863 | 6.8787 | 5.1141 |
| | $k$ | 1.0033 | 0.7371 | 0.9744 |
| Model 4 | $u$ | 2.3703 | 7.9240 | 0.0507 |
| | $\sigma$ | 7.8431 | 4.7796 | 5.7511 |
| | $k$ | 0.3833 | 0.2249 | 0.4785 |

The parameter $u$ represents the mean and helps indicate the position of the maximum number of cases. The parameter $\sigma$ determines the variance in terms of the Gaussian function and represents the duration of the outbreak. The constant $k$ is multiplied by the function and gives the size of the outbreak. It indicates the importance of temperature-dependent parameters.

that determines the parameter $\beta_{ex}$. These parameters (i.e., $u$, $\sigma$, and $k$) define the shape of $\beta_{ex}$, which is then multiplied by $b$ and $b_{hv}$ to correctly model an outbreak.

## 3.3 Outbreak model analysis

Next, an exhaustive analysis was performed for all years with outbreaks. In Comas, dengue cases were observed in 2017 and 2020. In Lurigancho, cases were observed in 2017 and 2019. In Puente de Piedra, cases were observed in 2017. Lima had no dengue cases during the study period with the exception of these three districts. Because not many dengue cases were reported, it was difficult to capture the infectious curve with traditional models, as shown in Fig 6.

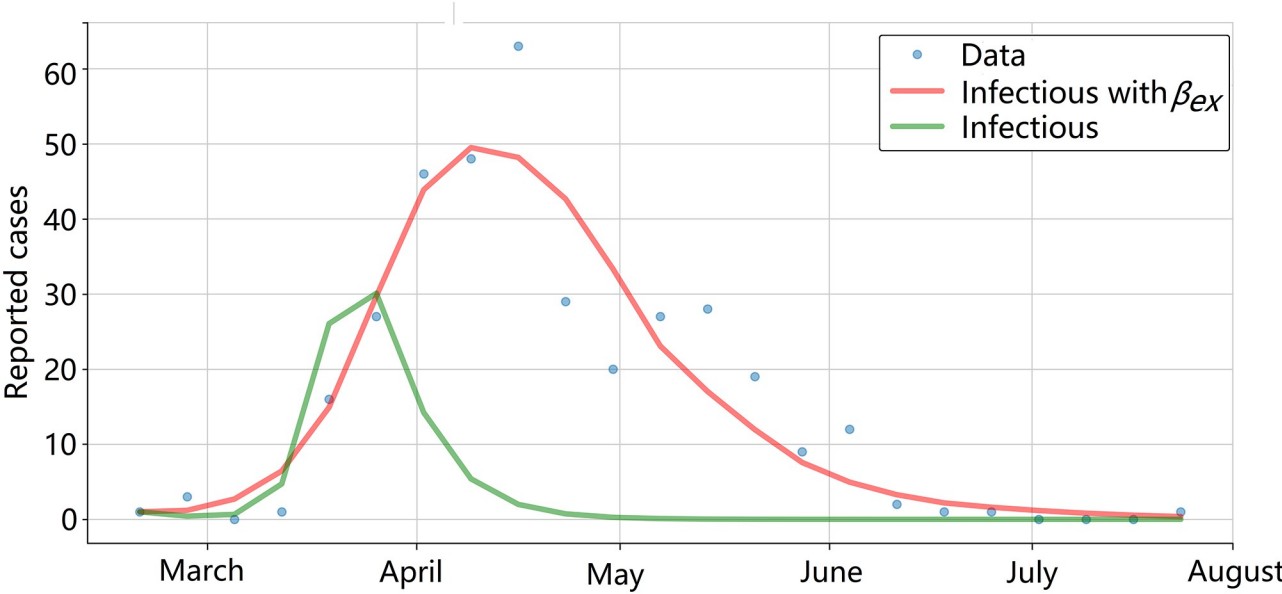

**Fig 6. SIR-SI model with the Gaussian exogenous variable and climatic conditions adjusted for 2017.** The adjusted analysis without the exogenous variable is in green, and the adjusted analysis with model 4 is in red.

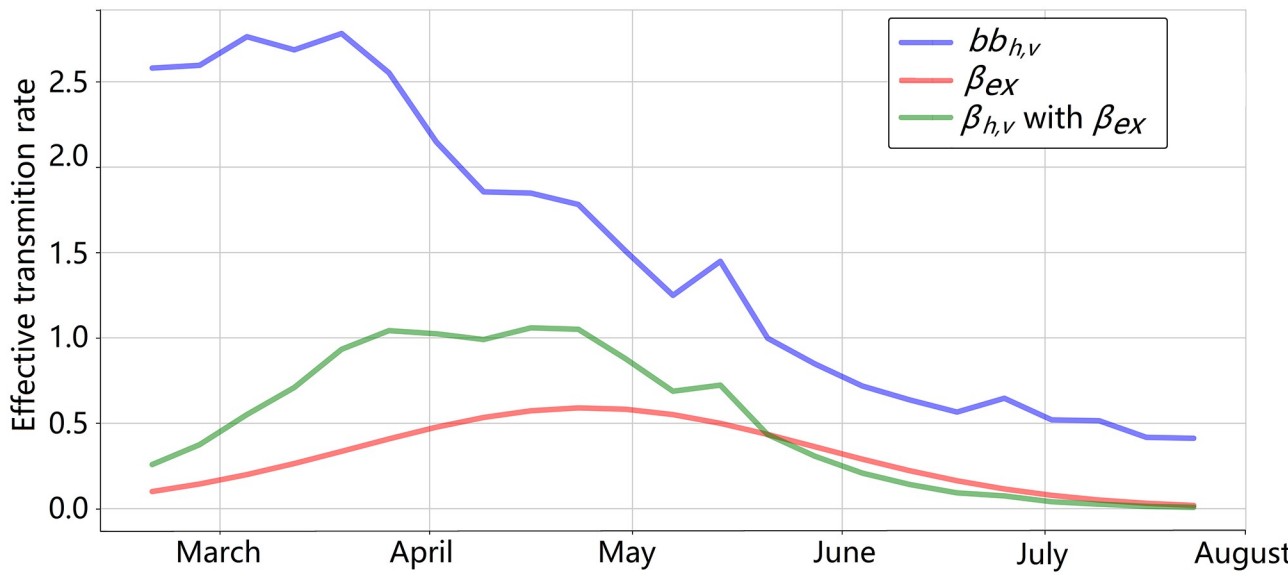

**Fig 7. Curves of $\beta_{ex}$, $\beta_{hv}$ based only on temperature, and $\beta_{hv}$ with $\beta_{ex}$ for 2017.**

**3.3.1 Districts.** Models 1–4 were applied to each district. Table 5 summarizes the model parameters.

Table 6 presents the model performances according to the evaluation metrics. The models were evaluated according to the AIC, BIC, and MLE. A better fit to the data was indicated by a lower value for an evaluation metric. In almost all cases, model 4 performed the best, followed by model 3. The difference between models 3 and 4 was not substantial. In the case of Puente de Piedra, model 3 actually fit the data better. This is because Puente de Piedra had few cases even at the peak of its outbreak.

**3.3.2 Analysis of 2017.** Fig 8 shows the dengue cases for 2017. The parameter $k$ defines the size of the curve and provides information about the importance of the weather to the outbreak. In Puente Piedra, the outbreak was mainly unrelated to the climate. In the other districts, the climate defined the magnitude of the outbreak because $k$ had values close to 1. The parameter $\sigma$ represents the duration of the outbreak and can be multiplied with climatic variables to obtain an outbreak correction factor, as shown in Fig 10. The model showed the best adjustment to the duration of the outbreak in Lurigancho. The parameter $u$ defines when the

**Table 5. Model parameters by district.**

| Models | Parameters | Comas | | Lurigancho | | Puente de Piedra |
|---|---|---|---|---|---|---|
| | | 2017 | 2020 | 2017 | 2019 | 2017 |
| Model 1 | $u$ | 5.6318 | 5.7444 | 7.9579 | 6.9908 | 8.1685 |
| Model 2 | $u$ | 5.263 | 5.3144 | 8.999 | 7.9664 | 6.6495 |
| | $\sigma$ | 0.8853 | 0.851 | 0.8328 | 6.8733 | 6.7153 |
| Model 3 | $u$ | 6.3313 | 5.1289 | 8.9719 | 6.8733 | 6.7153 |
| | $k$ | 5.9215 | 0.8744 | 5.1818 | 0.7334 | 3.4598 |
| Model 4 | $u$ | 2.9117 | 0.0252 | 8.9439 | 8.0206 | 5.6332 |
| | $\sigma$ | 7.2122 | 5.4298 | 2.6858 | 4.8441 | 0.4456 |
| | $k$ | 0.3298 | 0.4264 | 0.4205 | 0.2273 | 1.7437 |

**Table 6. AIC, BIC, and MLE values for each model.**

| Model | Metric | Comas | | Lurigancho | | Puente de Piedra |
|---|---|---|---|---|---|---|
| | | 2017 | 2020 | 2017 | 2019 | 2017 |
| Model 1 | AIC | 1821,7179 | 1935,5591 | 2636,6932 | 2793,8363 | 2684,7966 |
| | BIC | 1822,0205 | 1935,8617 | 2636,9958 | 2794,1389 | 2685,0992 |
| | MLE | 909,8589 | 966,7795 | 1317,3466 | 1395,9181 | 1341,3983 |
| Model 2 | AIC | 1509,0782 | 1678,8058 | 3009,8984 | 2381,3267 | 1078,8821 |
| | BIC | 1509,6833 | 1679,4109 | 3010,5036 | 2381,9319 | 1079,4872 |
| | MLE | 752,5391 | 837,4029 | 1502,9492 | 1188,6633 | 537,4410 |
| Model 3 | AIC | 945,2574 | 1583,6408 | 2213,2591 | 2022,8461 | 576,3646* |
| | BIC | 944,6522 | 1584,2460 | 2213,8643 | 2023,4512 | 576,9698* |
| | MLE | 470,3261 | 789,8204 | 1104,6295 | 1009,4230 | 286,1823* |
| Model 4 | AIC | 507,9966* | 413,7189* | 953,2745* | 1232,9001* | 722,7935 |
| | BIC | 508,9043* | 414,6266* | 954,1822* | 1233,8079* | 723,7013 |
| | MLE | 250,9983* | 203,8594* | 473,6372* | 613,4501* | 358,3967 |

Abbreviations: AIC, Akaike Information Criterion; BIC, Bayes Information Criterion; MLE, Maximum Likelihood Estimation. The superscript "*" denotes the best model.

peak of an outbreak occurs and its magnitude. In Comas and Lurigancho, the outbreaks peaked in the first week of April. In Puente Piedra, it peaked in the first fortnight of April. Observe that high values of $u$ and $k$ indicate how much the maximum value of the peak needs to be adjusted. For instance, in the case of Puente de Piedra, $\beta$ needed an adjustment of approximately 12 to reach the maximum peak. Without the adjustment of $\beta$, there would be no peak.

**3.3.3 Analysis of 2019.** Fig 9 shows (A) the incidence rate of dengue cases in Comas and (B) a comparison between the observed data and adjusted curve. The adjusted curve was obtained by using model 4 with $\beta_{ex}$ for 2019. Fig 9(C) and 9(D) show the corresponding results for Lurigancho in 2020. Fig 9(E) shows the values of $k$, $\sigma$, and $u$ for Comas and Lurigancho.

The results in Figs 8 and 9 demonstrate that model 4 with $\beta_{ex}$ underestimated the peak of the outbreak but adequately captured the beginning and ending of the outbreak in all cases. The latter is an important property because this denotes that the model successfully captured the complex phenomena of the outbreak despite the small amount of data.

## 3.4 Behavior of the infection rate $\beta$

Fig 10 shows the values of the temperature, $bb_v$, $\beta_{hv}$, $\beta_{ex}$, and $\beta_{ex}bb_v$ for all outbreaks and districts. There was a strong correlation between the temperature and $bb_v$, but $\beta_{ex}$ was needed to adjust $\beta_{hv}$ (similar results were obtained for $\beta_{vh}$). The importance of considering the modulation is relevant because several studies have reported that the climate-based variables determine the prediction and fitting of the SIR-SI model [14, 18, 24].

Fig 9 shows the importance of considering more components in the infection rate $\beta$. This is included by using $\beta_{ex}$. Fig 10 shows that modulating $bb_v$ depending on the temperature through the $\beta_{ex}$ provides more precise values for $\beta_{hv}$ and $\beta_{vh}$.

It is important to note that the climate of Lima is characterized by low levels of rainfall and variable temperature according to region owing to the effects of the ocean and the Andes Mountains.

*Evolution of $R_t$.* Fig 11 shows the value of the estimated real-time reproduction number $R_t$, its corresponding 90% credible interval, and the transmissibility $\beta_{vh}$ of the model. $R_t$ and $\beta_{vh}$

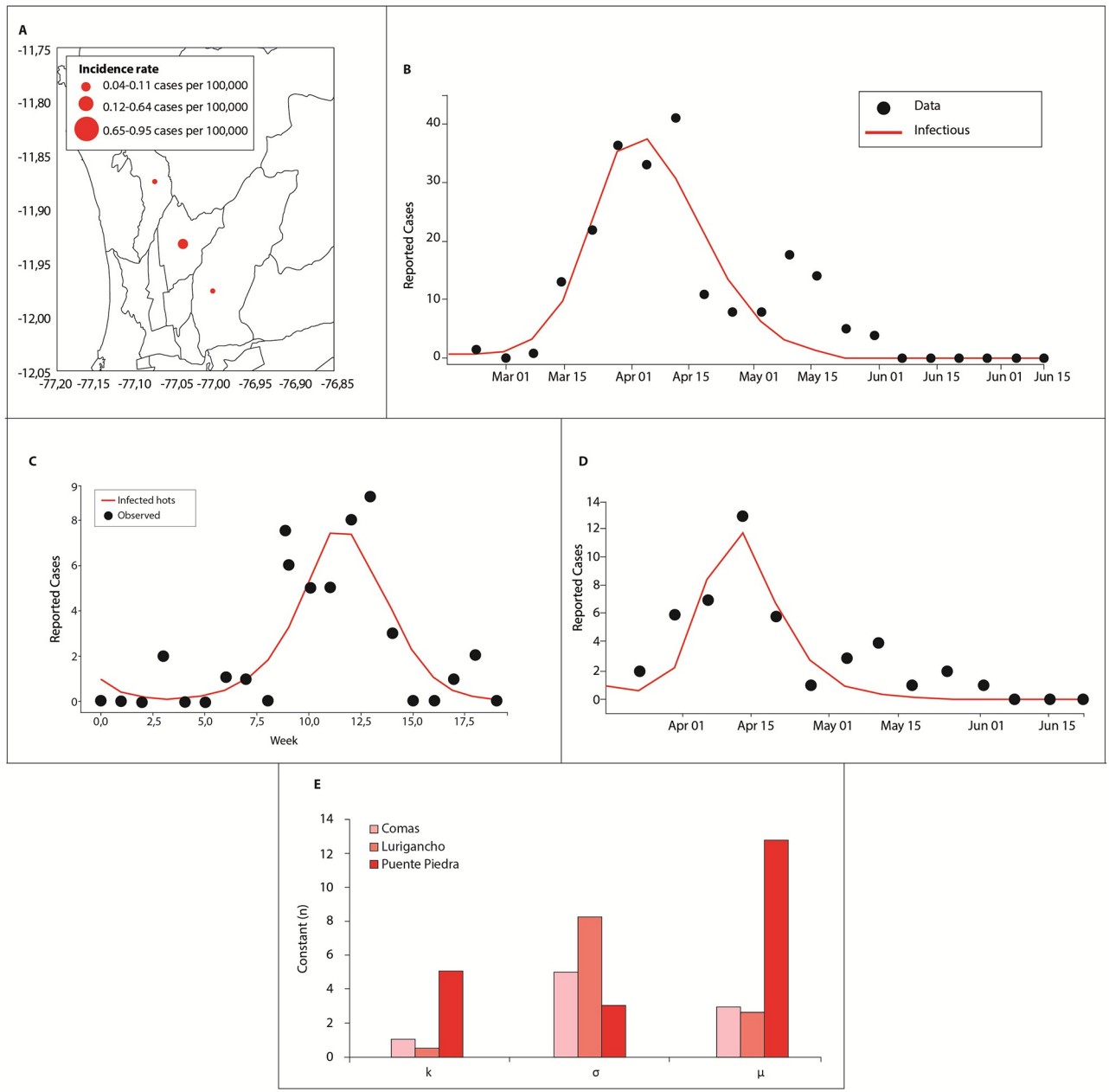

**Fig 8.** Distribution of dengue cases in Lima in 2017: (A) Incidence rate (per 100,000 inhabitants). Dengue cases obtained with model 4 for (B) Comas, (C) Lurigancho, and (D) Puente Piedra. The black dots and red line correspond to the reported dengue cases and adjusted curve using model 4, respectively. (E) Comparison between districts of the parameters $k$, $\sigma$, and $u$.

have a high correlation considering that the two estimated parameters were obtained by two different approaches. This is evidence that the proposed model captures the dynamics of an outbreak. A small and almost constant difference can be observed between the reproduction number and transmissibility. The plots also indicate that the reproduction number was greater than 1 a few weeks before the peak in reported cases.

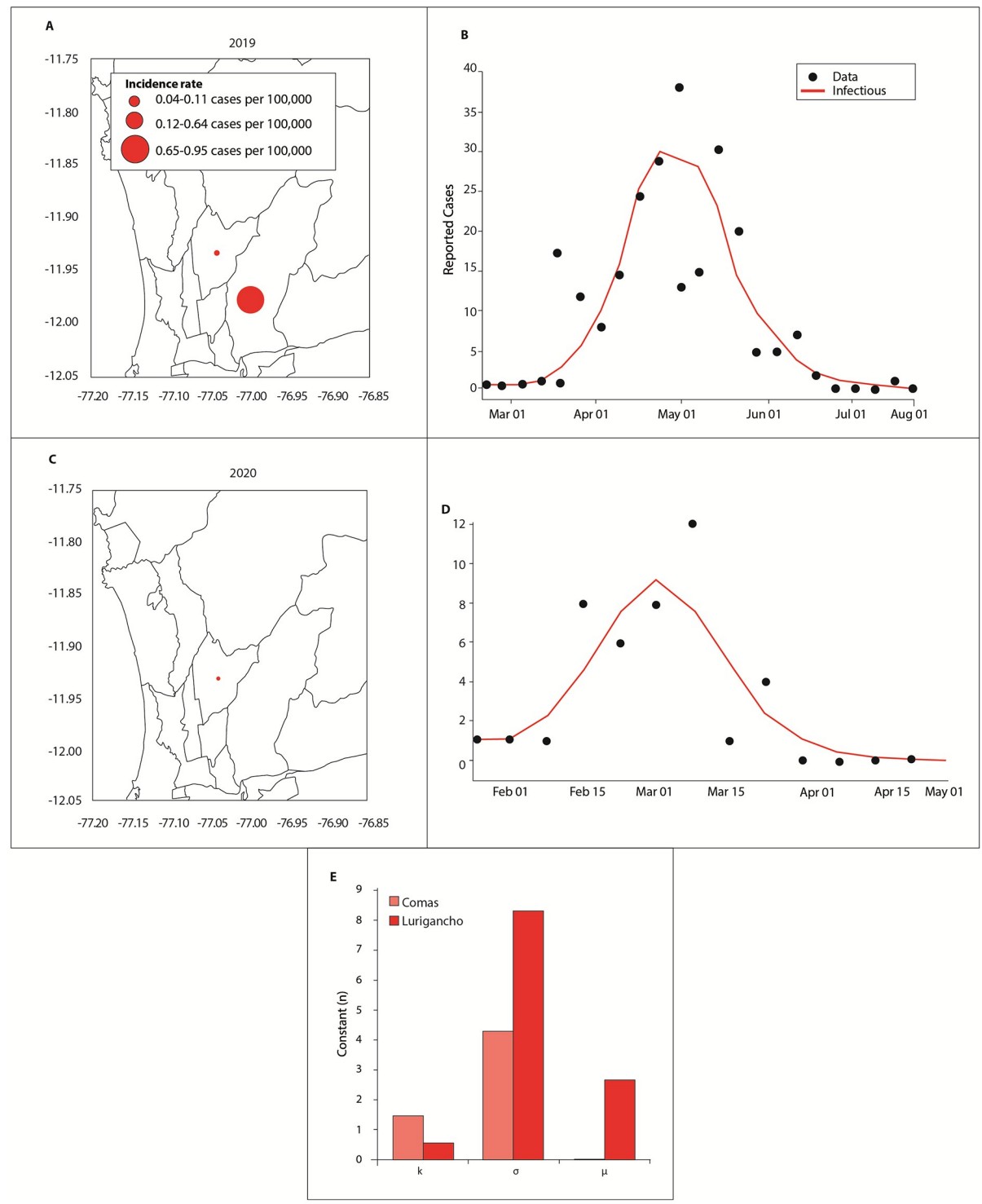

**Fig 9. Dengue cases in 2019 for Lurigancho and Puente de Piedra.** The black dots and red line correspond to the reported dengue cases and adjusted curve using model 4, respectively.

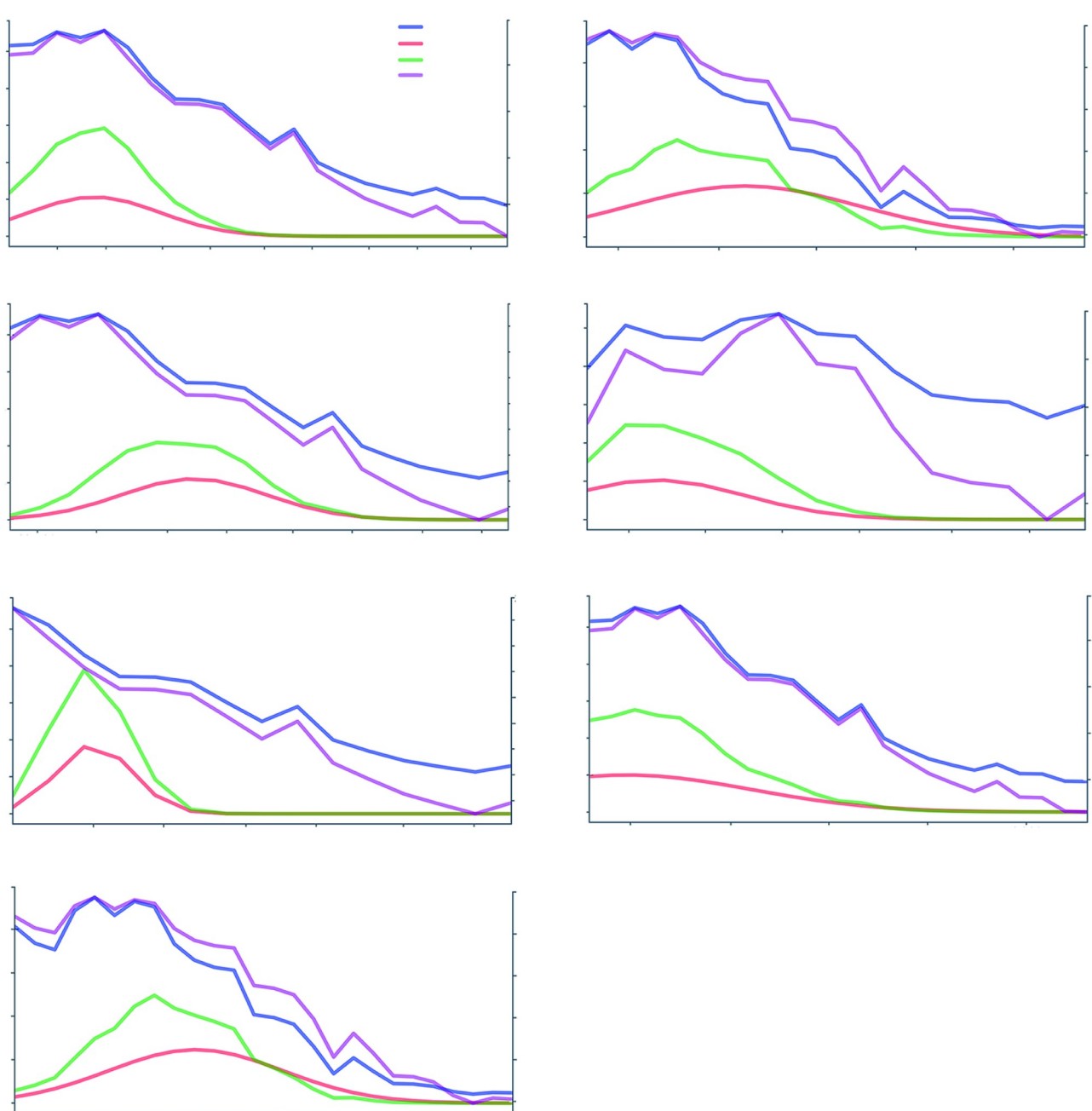

**Fig 10. Values of the temperature, $bb_v$, $\beta_{ex}$, and $\beta_{vh}$ (green) for outbreaks in 2017–2020 for Comas, Lurigancho, and Puente de Piedra.**

## 4 Discussion

Our results reveal the influence of weather on dengue transmission in Lima, Peru. The best-fitting model replicated the inter-annual variability of dengue cases in selected districts for 2017–2020. It is likely that the values of parameters change over time because the primary influencing factors that drive dengue transmission may change with the season or climatic conditions. Therefore, the ability of the SIR-SI model to resolve the variability of the annual case load and season length may be useful for various applications, such as studies focusing on the potential

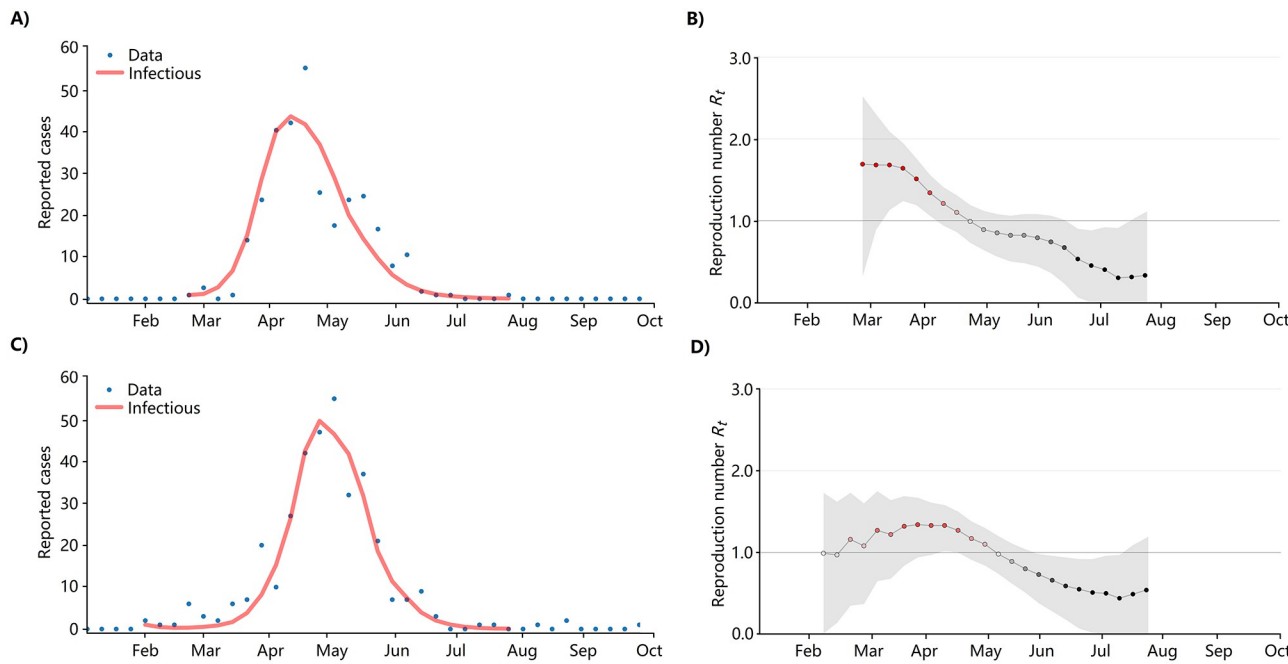

**Fig 11.** (Top) Reported cases (blue dots) and model prediction (Red). (Below) Values of the transmissibility $\beta_{vh}$ for Lima (including all districts) outbreaks in 2017 and 2020 (green lines) and the effective reproduction number $R_t$.

effects of climate change on dengue incidence and seasonality and other studies examining the causality of seasonal trends in relation to case numbers. Such a model can also be used for short-term predictions where parameter values are selected based on currently available case data and then simulations are run for forthcoming weeks using weather forecast data. Alternatively, the model can be used to build a dataset of epidemic profiles based on possible scenarios that could occur given present conditions.

The onset of the dengue season and peak in 2017 were not simulated well even when parameters were optimized specifically for the year, which illustrates the sensitivity and complexity of the disease. Many or all components of the virus ecology are constantly changing, and their responses to external factors such as weather depend on the situation. Meteorological conditions may not have had a strong influence on intra-annual variability in 2017. Other factors that are not included in the model may have dominated transmission that year. These include changing patterns in herd immunity to the specific circulating dengue serotype(s), the introduction of a new variant of a serotype earlier in the season, the implementation of intervention methods such as source reduction of habitats, or other human-related factors such as extensive use or reduction of water storage. While it was beyond the scope of this study to determine which of these factors may have influenced the transmission in 2017, a variant of one of the four serotypes could have been introduced early in the season, but all four serotypes had been circulating previously in Peru. Shifting herd immunity may play a role in reducing the overall level of reported cases but should not greatly influence the intra-annual variability of reported cases. Additionally, if higher levels of herd immunity played a role, a delay in the onset of cases would be expected, but we observed that reported cases peaked much earlier than the modeled cases. Given the high level of transmission in 2021, this early peak in 2017 may represent transmission propagated from the previous outbreak, where the initial transmission into the general population spread to a smaller adjacent geographical region. The

propagated transmission of dengue has been observed in other areas of the world. Changes in intervention strategies or patterns of container habitats may change the transmission dynamics by reducing or enabling transmission despite climatic conditions.

Although the temperature patterns are known to influence in dengue transmission in Peru, there is no information on the dengue transmission dynamics in Lima, Peru. Therefore, our findings have important implications for targeting mosquito control activities in poorly water serviced urban areas as Comas, Lurigancho, and Puente Piedra during the warm season. Our findings could also be useful for planning and targeting mosquito surveillance activities and preparing to the health centers for an increase in dengue cases or outbreak in the 3 study areas above all in poorly water serviced places. Our results also showed an analytical framework that successfully measured the dengue transmission dynamic in three districts of Lima with a limited number of cases. Therefore, our findings could be reproduced on a large scale in other areas spatially and temporally different from Lima with the necessary mathematical adjustments.

## 5 Conclusion

In summary, we explained the dengue transmission dynamics in Lima, Peru by using a SIR-SI model with climate-dependent parameters. Additional variables based on a Gaussian transmission rate were introduced to adequately capture the outbreak dynamics. These variables provide additional information about the duration of the epidemic ($\sigma$), the peak in the number of cases ($u$), and the influence of exogenous variables in the model ($k$). We also assessed the potential risk for dengue outbreaks via the vector capacity and intensity. We derived a formula for the reproduction number that qualitatively agreed with the Gaussian transmission rate introduced in the outbreak. The proposed model can be useful for analyzing the dengue transmission dynamics when few cases relative to the total population are reported. We observed good agreement between the collected data and model results when a Bayesian-Gaussian transmission rate was employed. The effect of climate was also observed, and a strong qualitative relationship was obtained between the transmission rate and the computed effective reproduction number $R_t$. This model incorporates an *ad hoc* mechanism to capture the processes involved in an epidemic. However, a question that remains for future work is an explanation for the internal outbreak process. Viable options include an entomological or transport-based explanation. In future work, we intend to incorporate these variables and compare them with the results of the present study.

## Supporting information

**S1 File.**
(PDF)

## Acknowledgments

The authors acknowledge the Universidad Tecnológica del Peru, the Centro de Investigaciones en Matemáticas, and the Universidad Nacional de Asunción

## Author Contributions

**Conceptualization:** Max Carlos Ramírez-Soto, Denisse Champin.

**Data curation:** Max Carlos Ramírez-Soto, Juan Vicente Bogado Machuca, Diego H. Stalder, Maria G. Mártinez-Fernández.

**Formal analysis:** Max Carlos Ramírez-Soto, Juan Vicente Bogado Machuca, Diego H. Stalder, Denisse Champin, Maria G. Mártinez-Fernández, Christian E. Schaerer.

**Funding acquisition:** Max Carlos Ramírez-Soto.

**Investigation:** Max Carlos Ramírez-Soto, Juan Vicente Bogado Machuca, Diego H. Stalder, Denisse Champin, Christian E. Schaerer.

**Methodology:** Max Carlos Ramírez-Soto, Juan Vicente Bogado Machuca, Diego H. Stalder, Maria G. Mártinez-Fernández, Christian E. Schaerer.

**Project administration:** Max Carlos Ramírez-Soto.

**Resources:** Denisse Champin.

**Software:** Juan Vicente Bogado Machuca, Diego H. Stalder, Maria G. Mártinez-Fernández, Christian E. Schaerer.

**Supervision:** Diego H. Stalder, Christian E. Schaerer.

**Validation:** Max Carlos Ramírez-Soto, Juan Vicente Bogado Machuca, Diego H. Stalder, Maria G. Mártinez-Fernández, Christian E. Schaerer.

**Visualization:** Max Carlos Ramírez-Soto, Denisse Champin, Christian E. Schaerer.

**Writing – original draft:** Max Carlos Ramírez-Soto, Juan Vicente Bogado Machuca, Diego H. Stalder, Denisse Champin, Maria G. Mártinez-Fernández, Christian E. Schaerer.

**Writing – review & editing:** Max Carlos Ramírez-Soto, Juan Vicente Bogado Machuca, Diego H. Stalder, Denisse Champin, Maria G. Mártinez-Fernández, Christian E. Schaerer.

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
