## [Decision Letter · Decision Letter 0]

14 Feb 2023

PONE-D-22-29852SIR-SI model with a Gaussian transmission rate Understanding the dynamics of dengue outbreaks in Lima, PeruPLOS ONE

Dear Dr. Ramírez-Soto,

Thank you for submitting your manuscript to PLOS ONE. It can be acceptable for publication after incorporating minor revisions based on the reviewer's comments.Therefore, we invite you to submit a revised version of the manuscript that addresses the points raised during the review process.

We look forward to receiving your revised manuscript.

Kind regards,

Jan Rychtář

Academic Editor

PLOS ONE

Journal Requirements:

Authors acknowledge the support given by P-2020-LIM-01, F: Universidad Tecnológica 434

del Peru, Lima, Peru. C.E.S. thanks CIMA. C.E.S. and D.S. acknowledge the support 435

given by PRONII - PROCIENCIA - CONACYT - FEEI

4. We note that Figure 1 in your submission contain map images which may be copyrighted. All PLOS content is published under the Creative Commons Attribution License (CC BY 4.0), which means that the manuscript, images, and Supporting Information files will be freely available online, and any third party is permitted to access, download, copy, distribute, and use these materials in any way, even commercially, with proper attribution. For these reasons, we cannot publish previously copyrighted maps or satellite images created using proprietary data, such as Google software (Google Maps, Street View, and Earth). For more information, see our copyright guidelines: http://journals.plos.org/plosone/s/licenses-and-copyright.

5. Please remove your figures from within your manuscript file, leaving only the individual TIFF/EPS image files, uploaded separately. These will be automatically included in the reviewers’ PDF.

Additional Editor Comments:

This is a well written manuscript that can be acceptable for publication after incorporating minor revisions based on the reviewer's comments.

Reviewers' comments:

Reviewer's Responses to Questions

**Comments to the Author**

1. Is the manuscript technically sound, and do the data support the conclusions?

Reviewer #1: Yes

2. Has the statistical analysis been performed appropriately and rigorously? 

Reviewer #1: Yes

3. Have the authors made all data underlying the findings in their manuscript fully available?

Reviewer #1: Yes

4. Is the manuscript presented in an intelligible fashion and written in standard English?

Reviewer #1: Yes

5. Review Comments to the Author

Reviewer #1: The manuscript presents a temperature-dependent SIR-SI compartmental model to understand the transmission dynamics of dengue in Lima Peru. The authors made an honest effort to achieve methodological soundness, with a thorough description of all the differential equations and assumptions that was utilized in the model.

The manuscript is well written and easy to read. In my opinion, one of the strengths of the manuscript is that it provides insights into dengue transmission in a contextualized low-transmission area (i.e. Lima), with its results/finding providing useful guide in the development of effective local control and mitigation strategies.

A general statement for improvement of the manuscript will be to reduce the number of equations that is presented in the main manuscript and emphasis the relevance of this work in real life public health implementation. Most of these equations and detailed explanation (e.g., equation 34 & 35) can be moved to a supplementary material.

Below are a few additional comments and questions that also need to be addressed to improve the methodological soundness of the paper.

1. Did the authors consider the egg-to-adult survival and development rate of the vector? As these mosquito traits relevant to transmission and respond strongly to temperature. If this was not considered, authors need to clearly state reasons and assumptions made in the model.

2. The manuscript was not clear on the starting conditions for the human population (which in turn affects the starting vector population). Table 1 states the population was estimated by the district, what was the population of each district? What this number varied for each year to reflect population change?

3. Based on the comment above, did the authors consider varying the initial mosquito populations based on the seasonal pattern of mosquitoes? If mosquito entomological surveillance data in available for Peru, this will provide better insight into seasonal pattern and population of the vector.

4. Table 1. Model parameters can the authors add the minimum, maximum and rate constant, of the parameters to this table, mostly for the temperature dependent parameters (somewhere stated in the equations, adding this to the table will aid readers understanding).

5. Lines 92-96 suggest that weekly temperature variation was considered for the modeling. My assumption is this was done to match the weekly epidemiological dengue data. Did the authors consider utilizing daily temperatures as oppose weekly? because in the real-world organisms do not typically experience constant temperature environments in nature for a week. Also, your model needs to be able to account for the fluctuations in daily temperature range.

6. PLOS authors have the option to publish the peer review history of their article (what does this mean?). If published, this will include your full peer review and any attached files.

Reviewer #1: **Yes: **Dr. Donald Salami

---

## [Author Response · Author response to Decision Letter 0]

26 Mar 2023

Response-to-reviewers: Manuscript PONE-D-22-29852 “SIR-SI model with a Gaussian transmission rate Understanding the dynamics of dengue outbreaks in Lima, Peru”

We thank the Reviewers and Editor for your comments and constructive criticism, we believe that the quality of our manuscript has been significantly improved. We have revised our paper in a point-by-point manner. 

Journal Requirements:

Comment 1. We note that the grant information you provided in the ‘Funding Information’ and ‘Financial Disclosure’ sections do not match. When you resubmit, please ensure that you provide the correct grant numbers for the awards you received for your study in the ‘Funding Information’ section. 

Response 1. Thank you for your comments. We performed the correction in submitted system. 

Comment 2. We note that Figure 1 in your submission contain map images which may be copyrighted. All PLOS content is published under the Creative Commons Attribution License (CC BY 4.0), which means that the manuscript, images, and Supporting Information files will be freely available online, and any third party is permitted to access, download, copy, distribute, and use these materials in any way, even commercially, with proper attribution. For these reasons, we cannot publish previously copyrighted maps or satellite images created using proprietary data, such as Google software (Google Maps, Street View, and Earth). For more information, see our copyright guidelines: http://journals.plos.org/plosone/s/licenses-and-copyright.

Response 2. Thank you for your comments. Figure 1 was generated with python scripts using open libraries and shape files. This figure was made by the authors.

Comment 3. Please remove your figures from within your manuscript file, leaving only the individual TIFF/EPS image files, uploaded separately. These will be automatically included in the reviewers’ PDF. 

Response 3. Thank you for your comment. Figures are removed and attached separately.

Comment 4. Please review your reference list to ensure that it is complete and correct. If you have cited papers that have been retracted, please include the rationale for doing so in the manuscript text, or remove these references and replace them with relevant current references. Any changes to the reference list should be mentioned in the rebuttal letter that accompanies your revised manuscript. If you need to cite a retracted article, indicate the article’s retracted status in the References list and also include a citation and full reference for the retraction notice.

Response. Thank you for your comments. NA

Additional Editor Comments:

This is a well written manuscript that can be acceptable for publication after incorporating minor revisions based on the reviewer's comments.

Response. Thank you for your comments. 

Comments to the Author

Reviewer #1: The manuscript presents a temperature-dependent SIR-SI compartmental model to understand the transmission dynamics of dengue in Lima Peru. The authors made an honest effort to achieve methodological soundness, with a thorough description of all the differential equations and assumptions that was utilized in the model. The manuscript is well written and easy to read. In my opinion, one of the strengths of the manuscript is that it provides insights into dengue transmission in a contextualized low-transmission area (i.e. Lima), with its results/finding providing useful guide in the development of effective local control and mitigation strategies.

Response. Thank you for your comments. 

Comment. A general statement for improvement of the manuscript will be to reduce the number of equations that is presented in the main manuscript and emphasis the relevance of this work in real life public health implementation. 

Response. Thank you for your comments. This paper has a component of data-driven modeling and capturing the phenomenology immersed in them. In this sense, the equations and assumptions are transparently displayed to ensure the work is reproducible. We re-arranged some parts of the article by creating a supplementary section with proofs and equations. We believe that this will simplify the lecture of the article and help the reader. Additionally, we include a paragraph on the implications for public health in the Discussion section. 

Comment. Most of these equations and detailed explanation (e.g., equation 34 & 35) can be moved to a supplementary material. 

Response. Thank you for your comments. We decided to move section 2.3.1, 2.3.1 y 2.3.2 to the suplementary material sections S1, S2 and S3. 

Below are a few additional comments and questions that also need to be addressed to improve the methodological soundness of the paper.

Comment 1. Did the authors consider the egg-to-adult survival and development rate of the vector? As these mosquito traits relevant to transmission and respond strongly to temperature. If this was not considered, authors need to clearly state reasons and assumptions made in the model.

Response 1. Thank you for your comments (Christian). The egg-to-adult survival and development rate of the vector was not considered, we consider that the carrying capacity of the system has not yet been reached and therefore there is no restriction in larval growth. The temperature effects were included in the parameters of Equations (5) and (6), e.g., the μ, b, bh the mosquitos mortality rate, bitting rate, and transmission probability per bite, respectively. This is assumption 4, whose explanation was improved so that temperature dependence is the main factor considered. 

Comment 2. The manuscript was not clear on the starting conditions for the human population (which in turn affects the starting vector population). Table 1 states the population was estimated by the district, what was the population of each district? What this number varied for each year to reflect population change?

Response 2. Thank you for your comments. Incluir en la limitación del estudio (Christian). La población fue para cada distrito. Considerando una población constante. Tabla 1. 

Response 2. Thank you for your comments. The starting human population is a variable in our model. This is due to consider the lack of information, but also to deal with the small cases in a region with variable density in comparison with the range of the mosquitoes flight. The population was considered for each district as mentioned in Table 1: "Variable Nh0 according to each district", "Estimated". Hence, our model fits Nh0 for each year (53 weeks) and for each district. Hence, we assume that the population of susceptible humans is limited by the radius of action of the mosquito, so we leave it as a parameter to be estimated. For each year, we assume that the population is constant.

Comment 3. Based on the comment above, did the authors consider varying the initial mosquito populations based on the seasonal pattern of mosquitoes? If mosquito entomological surveillance data in available for Peru, this will provide better insight into seasonal pattern and population of the vector.

Response 3. Thank you for your comments. As mentioned above, our model is the estimates human population, and with this value, we approximate the mosquito population by assuming that the Nv0=2Nh0 (see Lee et al. 2018 ). The seasonal patterns are introduced in the model throughout the temperature-dependent parameters(see response 1 above). 

Comment 4. Table 1. Model parameters can the authors add the minimum, maximum and rate constant, of the parameters to this table, mostly for the temperature-dependent parameters (somewhere stated in the equations, adding this to the table will aid readers understanding).

Response 4. Thank you for your comments. In Table 1, we introduce references to Figures 4 and 7, where the reader can verify the range and shape of the temperature-dependent parameters. 

Comment 5. Lines 92-96 suggest that weekly temperature variation was considered for the modeling. My assumption is this was done to match the weekly epidemiological dengue data. Did the authors consider utilizing daily temperatures as oppose weekly? because in the real-world organisms do not typically experience constant temperature environments in nature for a week. Also, your model needs to be able to account for the fluctuations in daily temperature range.

Response 5. Thank you for your comments. Because the number of cases is low and the population size of the district is limited, making a daily distribution of the time series would be very noisy, because is possible to have unreported cases. In addition, the time period between the occurrence of cases and registration in the notification system may have delays. Another limitation is that the notification of dengue cases by the Peruvian dengue surveillance center is weekly. So, we restrict our study to a weekly scale. However, daily and hourly fluctuations in temperature could be considered to integrate the mosquito vector equations, as vector dynamics are different from humans over time. We discuss this limitation in Section 2.1, where the dataset is presented.

---

## [Editor Report · Decision Letter 1]

28 Mar 2023

SIR-SI model with a Gaussian transmission rate Understanding the dynamics of dengue outbreaks in Lima, Peru

PONE-D-22-29852R1

Dear Dr. Ramírez-Soto,

We’re pleased to inform you that your manuscript has been judged scientifically suitable for publication and will be formally accepted for publication once it meets all outstanding technical requirements.

Kind regards,

Jan Rychtář

Academic Editor

PLOS ONE

Additional Editor Comments (optional):

Thank you for adequately incorporating all comments. The paper is now acceptable for publication.

---

## [Editor Report · Acceptance letter]

4 Apr 2023

PONE-D-22-29852R1 

SIR-SI model with a Gaussian transmission rate: Understanding the dynamics of dengue outbreaks in Lima, Peru  

Dear Dr. Ramírez-Soto:

I'm pleased to inform you that your manuscript has been deemed suitable for publication in PLOS ONE. Congratulations! Your manuscript is now with our production department. 

Kind regards, 

on behalf of

Dr. Jan Rychtář 

Academic Editor

PLOS ONE